# Quodlibet with Meninas

**Maria Gil Ulldemolins** 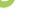

Faculty of Architecture and Arts, Hasselt University, 3590 Dipenbeek, Belgium; maria.gilulldemolins@uhasselt.be

**Abstract:** In Diagrammatic Writing (2013), Johanna Drucker discusses the power dynamics between texts interacting on a page. So-called autotheoretical texts often engage in similar types of performative and relational lay-outs, and yet, not much has been written about this formal phenomenon. Bearing this in mind, I propose an experiment that performs relations by thinking with, and through, Las Meninas, a self-portrait that is not strictly about the self. All that surrounds Velázquez in the painting (the work-in-progress we do not see, the ensemble of courtly characters, the framed reproductions of masters' works, the much-discussed mirror reflection) informs and contextualises the portrait, but also explodes it into much more. This paper thus attempts to ask whether autotheory can, by being aware of performative and diagrammatic writing, together with the use of images as citations, decentralise the auto- and become a more choral scene, a cluster, a textual quodlibet or medley. Can a form of writing make space for a multitude, or even, a multitude into a space? Can the autotheoretical self be only one more of many characters, present, with agency, but off-centred?

**Keywords:** Meninas; autotheory; diagrammatic writing; performative writing; citationality; quodlibet

## 1. The Beholder
### 1.1. The Girls

There are three little girls (Figure 1):

Margarita: A five-year-old princess in 1656, a blonde thing dressed in thick, white satin that makes a frou-frou sound just from looking at it.

Lucy: three or four, bursting into the 1980s in an ample dress, sitting on an indoor swing hung from a door frame, beaming in a chunk of sunshine.

Maria: similarly aged. It's not 1990 yet. Unlike the other two, she is not wearing much, just high-waisted knickers and puppy fat.

The three girls' left hands reach out to the world that is most immediately accessible to them: the voluminous skirt, the swing's chain, the fleshy top of one thigh. A still, soft notion of personal space—the space needed for one's presence, for one's steadiness and movement, the early reassurance (or damnation) that one's body holds the self.

The first one is a painting; the second, a painting of a photograph; and the third, a photograph.

> (None of these girls exist anymore. This text is not really about them.)

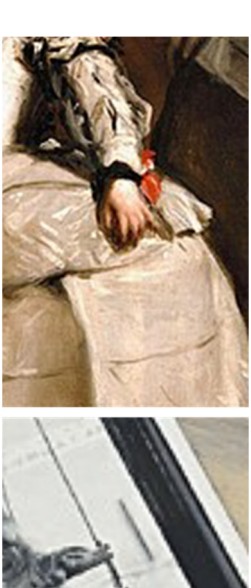

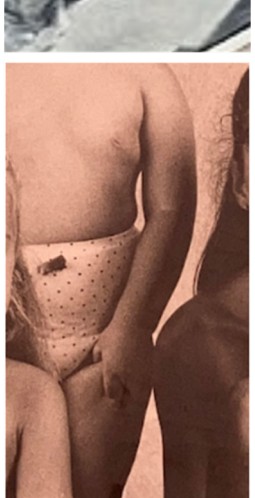

**Figure 1.** The girls' left hands, details (**Top**) (Velázquez 1656). (**Middle**) (McKenzie 2021). (**Bottom**) (Author's own picture (2021) of printed ad for the author's family business, circa 1989).

*1.2. The Gesture[1]*

The term autotheory 'refers to the integration of theory and philosophy with autobiography, the body, and other so-called personal and explicitly subjective modes' (Fournier 2021, p. 7). Reminiscent of a 'memoir with footnotes' (p. 7), it often 'inscribes a performative mode of citation' (p. 7).

The performative aspect cannot be underestimated. Performative language is to be understood in the J. L. Austin sense, as language that can *perform* an act, have a tangible effect merely by being uttered. It happens all around us every day, and the most classic examples are contract-like: vows, promises.

> (Autotheory, in a way, carries an implicit promise: a tangible self, essential to the text. I would describe this expectation and commitment as performative, yes.)

From language we can stretch the concept into performative writing, of which Della Pollock says: 'writing as *doing* displaces writing as meaning; writing becomes meaningful

in the material, dis/continuous act of writing . . . writing becomes itself, becomes its own means and ends, recovering to itself the force of action.' (p. 75)

The action, as Pollock describes it, seems to enclose writing into its own world: the text becomes both its own site and agent. Although she specifies that we should not get distracted by the playful formality of this business ('Performative writing is, for me, precisely not a matter of formal style (especially in the degraded sense of glinting, surface play)' (p. 75)); that is, I am afraid, precisely where I am going.

<div align="right">(left hand pinching own left thigh)</div>

Going back to the idea of performative citation, and bringing it all together, the citational aspect can similarly 'become its own means and ends', and a 'force of action,' too. The act of folding fistfuls of other's words and thoughts into your own not only serves a purpose in the building of the discourse, but is a self-defining gesture in itself, a flourish, or a curtsy.

<div align="right">(a hand resting on expensive, soft materials)<br>(this, you understand, is my own left hand,<br>trying to grab onto a process of becoming, trying<br>to feel its risks, limitations, and augmentations[2])</div>

### 1.3. The Moon

There is a type of crime that consists of crashing a large vehicle against a façade as a way to get access to a space. In Spanish, it's called *alunizaje*, moon-landing.[3]

<div align="right">(Consider this the methodological bit none of us<br>likes to write.)</div>

1.  We make a text that is a vehicle,
2.  we crash it against a surface that is art history,
3.  we scurry away taking handfuls of whatever shines brighter, whatever tickles one's fancy.

What becomes the work's red thread, its conceptual glue, is a blend of desire and opportunity: whatever you get to grab and drive away with before alarms and sirens cut the air, dust blooming into what used to be a wall, and into your lungs.

### 1.4. The Pleasure

A quodlibet is a medley. The term must be useful as a cultural form, for it stubbornly carves space for itself in different disciplines and mediums.

In music, it refers to harmonisation exercises, often parodic, that peaked in popularity in the 15th and 16th centuries (Kuiper 2016). These were ingenious combinations of well-known and/or smutty tunes, played or sung together to the audience's delight (hence, the quodlibet name, which is the Latin for 'whatever pleases'). This practice, which we would now call a 'mashup', is fantastically well-established, having been used from Bach to Nina Simone (Neely 2016), and is practically everywhere in contemporary music.

In mediaeval philosophy, it refers to a debate prompted by a member of the public, often inspired by current hot topics (Sweeney 2019). It is not harmonies that are strung together here, but thoughts, prompted by others's suggestions, an improvised debate.

In painting, it designates a trompe l'oeil of a table or board surface, littered with things such as papers (notes, letters) and other typical studio paraphernalia (writing utensils, palettes); and even sand clocks, or the odd musical instrument (violins, lutes) (Oxford Reference Editors 2006) (Figure 2).

<div align="right">(The painter Lucy Mckenzie is possibly the<br>best-known contemporary maker of such artefacts.<br>She was once, if I am not mistaken, also one of the girls<br>mentioned earlier.)</div>

The quodlibet gives historic weight to a gesture that betrays attraction, straying fingers.

It reminds me of the architecture of fragmentary autotheory, of how I write. Written quotes falling into strategically placed handbags, like drugstore lip glosses, glittering under the cold fluorescent light.

(I used to be one of the three little girls, but I was never one of *those* girls. First daughter, Catholic school, A type, you get the picture.)

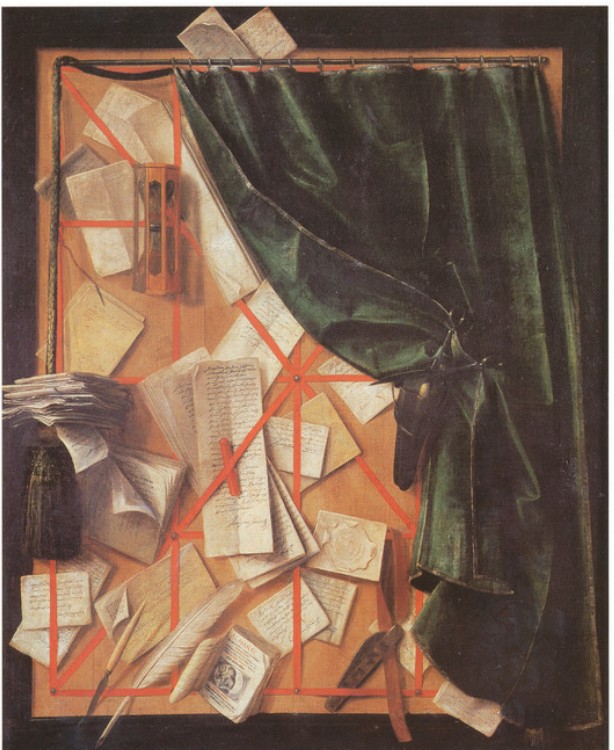

**Figure 2.** Trompe l'oeil of a quodlibet with hourglass, razor, scissors and curtain (Gijsbrechts 1664).

*1.5. The Writing*

(with my apologies to Pollock)

Joanna Drucker's *Diagrammatic Writing*'s last sentence is 'This is a book that is as close as possible to being entirely about itself.' (p. 32) *Diagrammatic Writing* gives textual devices a form of visual self-awareness and self-referentiality. For instance, a bit of marginalia written in tiny cursives, pushing the body of the text from the left, reads: 'Marginalia are the gadflies of discourse, the directives, the instructions on reading and the goad to critical thought. They enter the page like darts, small interventions, pebbles on the road.' (p. 20)

(Here, I am not so much the body as the gadfly; nibbling in the hope of an embodied critique, bothersome, dubitative; a test of one's own existence and position.)

Drucker's book tries to decipher the relations of power between the different textual parts and a text as a whole, such as how 'A larger point size makes a claim for the authority' (p. 15); or how:

Entanglement is less hierarchical than embedment.

*In a condition of entanglement, one text does not have to be smaller*

One example of entanglement is interlinear discussion

*than the other, and when it is, then the sense of its secondariness is*

and commentary. This might take the form of dialogue,

> *immediately established. This text might suggest that the role of*

refutation, objection, agreement, expansion, extension,

> *entanglement is mere commentary, exposition without any relation*

or any of an infinite number of other positions.

> *to the text into which it is inserted. However, the opposite is true.*

Entanglement complicates a text.

> *This text makes it difficult to read the first text on its own.*

(p. 17, formatting in the original).

The formal play, the way writing makes itself happen, feels, of course, performative in itself, despite's Pollock's earlier warning.

> (One of autotheory's troubles, one could argue, is the entangled self, risking a debacle of proportion, turning one of the parts into 'mere commentary', or the appearance of it, even if 'the opposite is true.')

### 1.6. The Codex

Drucker's position regarding the performativity of diagrammatic texts is clear: they go hand in hand. In 'Diagrammatic and Stochastic Writing and Poetics' she states that diagrams '*work*—they *do* something rather than *represent* something' (Drucker 2014b, p. 125, emphasis in original). They *perform*. In her lecture 'Diagrammatic Form and Performative Materiality', Drucker advocates for the 'performative materiality' of diagrammatic instruments to be regarded as 'generative'. These tools are capable of knowledge production, and more specifically, of articulating semantic values through the visualization of a logic of relations (as in the extract shown above). This means that the diagram, or the diagrammatic text ('the codex book, the page, the graphical organization of layout on the screen—these are all diagrammatic formats' (Drucker 2014b, p. 125)), is a *structure* (Drucker 2014a). And as such, it is dynamic, capable of escaping a 'one-on-one' relationship with its concern (Drucker 2014a).

Once the relational and structural potential is on the table, the jump towards a *spatial* potential is right there: 'a book is not a static object but a dynamic space, . . . an organized arrangement of elements whose spatial relations encode semantic value' (Drucker 2014b, p. 123).

We are right at the palace's doors, text to space, text to painting, but still, concerned with dynamics of being and becoming, within oneself, within one's tradition: 'a book is always an argument about what a book is (as object, text, discourse), as surely as any novel, poem, painting, sculpture, performance always proceeds from an assumption about what form performs in its initial coming-into-being' (Drucker 2014b, pp. 122–23).

### 1.7. The Palace

In Diego Velázquez's *Las Meninas*, 1656 (Figure 3, left), we see a room on the ground floor of King Philip IV's Alcázar palace in Madrid. It is large, with high ceilings, and suitably somber.

> (There used to be a bit of text, here, about the Spanish art of domestic light regulation. About how it is one of the things men do around the house, control and care combined. It even declared that indoors, all Spanish men are vampires—because of the light sensitive thing, naturally. I have a thing for parentheticals that trail off, surely a performative fixation, too. But sometimes they get out of hand. Open windows that need to be blinded. Like cyclops (see, again). Like a Spanish interior, which is, I suppose, what I am.)

To the right of the painting, you notice the row of windows, and they are shut, as they should be, except for the closest one to the viewer, which allows enough light in to bathe the whole scene (including the moon-like princess, bouncing off the sunlight all by her stiffly skirted self).

> 'Is this outlier a rogue text?
> Or is it related to the others?
> How should it be considered?'
> (Drucker, p. 9, formatting in the original)

I have nothing particularly insightful to say about this room (other than introducing it to Drucker's work), or the characters depicted in it. Western art history is littered with discourse about their particularities. But I would like to borrow it so I can think with it, through its gestures, in the in-betweens, as a self-portrait-as-many.

> (And by borrowing I mean grabbing its surface. And by thinking with it, I mean running the auto-theoretical van right against it.)

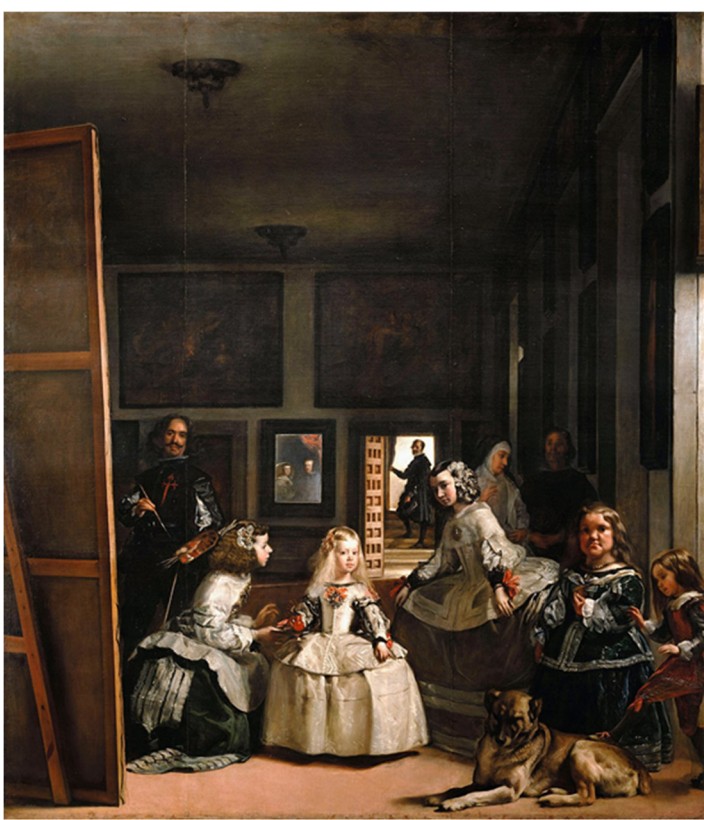 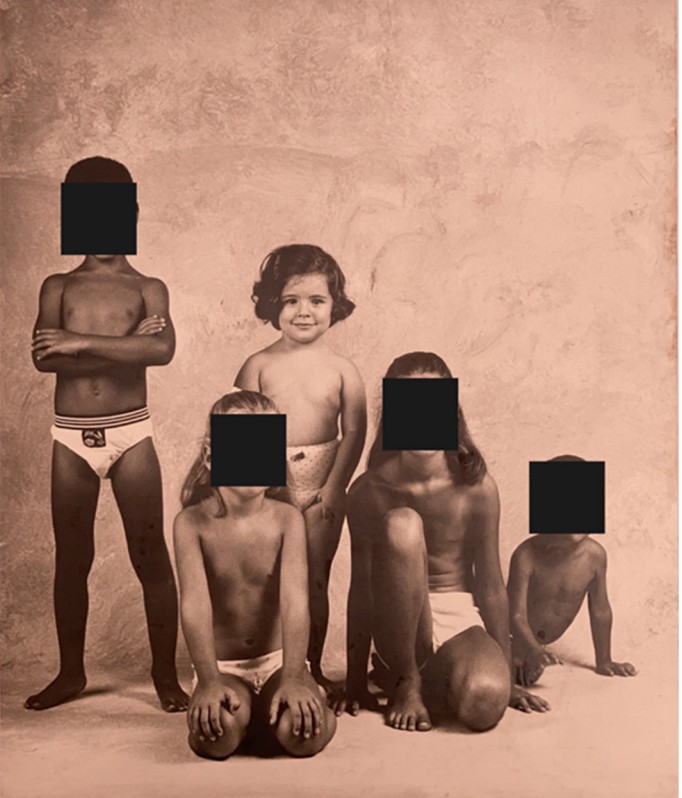

**Figure 3.** (**Left**) Diego Velázquez. 1656. *Las Meninas*. (**Right**) Author's own picture of ad campaign for the author's family's clothing store, c. 1989. Photo modified by author.

## 2. The Artist

### 2.1. The Lovables

In *The coming community*, Giorgio Agamben picks at the 'adjective' *quodlibet*:

The common translation of this term as "whatever" in the sense of "it does not matter which, indifferently" is certainly correct, but in its form the Latin says exactly the opposite: *Quodlibet ens* is not "being, it does not matter which," but rather "being such that it always matters." The Latin always already contains, that is, a reference to the will (*libet*). Whatever being has an original relation to desire. (Agamben 2013, p. 1)

(A quick reminder that to be desired is to matter (or, at
least, to be worthy of moon-landing). And in
autotheory the self matters, veins of desire creating
blue-green cartographies that turn red when cut.)

For Agamben, quodlibet is related to a form of singularity, to what he calls being-*such*
(italics in the original). This being-*such* escapes the trappings of identifying with a larger
group, and allows the subject to be no less and no more than the collection of their own
particularities, and appreciated for them: 'The lover wants the loved one *with all of its
predicates*, its being such as it is'. (p. 2, emphasis in the original)

This understanding of singularity transforms the quodlibet into the Lovable (capital
L), which is in turn intelligible, understandable, pick-apart-able. A compendium of itself,
capable, in an erotic fashion, to transport, not elsewhere (ecstatically), but 'toward its own
taking-place—toward the Idea' (p. 2).

A being that contains its own genesis and source. Something like Robert Morris' *Box
with the Sound of Its Own Making*, 1961[4]. Or Pollock's performative writings.

(Rustle, rustle. Bang, bang, bang.

Typing clatter.)

### 2.2. The In-Between (Figure 4)

'Juxtaposition pretends to
parity. In actuality, the urge
to competition belies this
illusion. A strong tension
between either/or struggles
with the equally developed
impulse towards both/and.'

'Juxtaposition pretends to
parity. The sheer force
of the space between in
relation to the condition
of alignment creates an un-
resolvable situation. These
are not the same text.'
(Drucker 2013, p. 14, format in the original).

*Diagrammatic Writing* is a spatial affair, especially when it comes to in-betweens,
interstices, as above, charging the white path between columns with 'sheer force', and a
'strong tension'. These gaps between texts establish (with different degrees of violence)
various relations between pieces of texts, which then make these fragments into a body,
or a whole other type of space, a world with which to engage: 'The relation of one text
to another and each text to many and all others inside a work and outside creates a fully
entangled field.' (p. 17)

(Which also means, of course, as we've seen earlier,
that it also becomes more complicated, riskier, too. The
entanglement binds the self to the world and vice
versa. Someone's daughter stands in the middle of a
crowd, aware of expectations but unable to affect them,
a gentle dog nodding off in the foreground.)

Drucker's notion of entanglement also adds a sense of *inter*face to the *surf*ace. I
particularly love how she differentiates 'wordspace from worldspace' (p. 12), which
assigns importance, or even agency, to both: they are both positive spaces, active, activable.
Drucker's entanglement reveals different hierarchies and roles within the text, and with it,
the architectural thresholds between them. It then allows for the field to be *expanded,* the
structural relations between paragraphs and pages turned into an on-going gesture: 'the
associative field within the text creates endless opportunities for branching or/breaking the
line to follow lines of thought.' (p. 11) I would argue this is very close to Pollock's notion
of a 'nervous' (p. 90) text, as in nerve-like, sparking, non-linear.

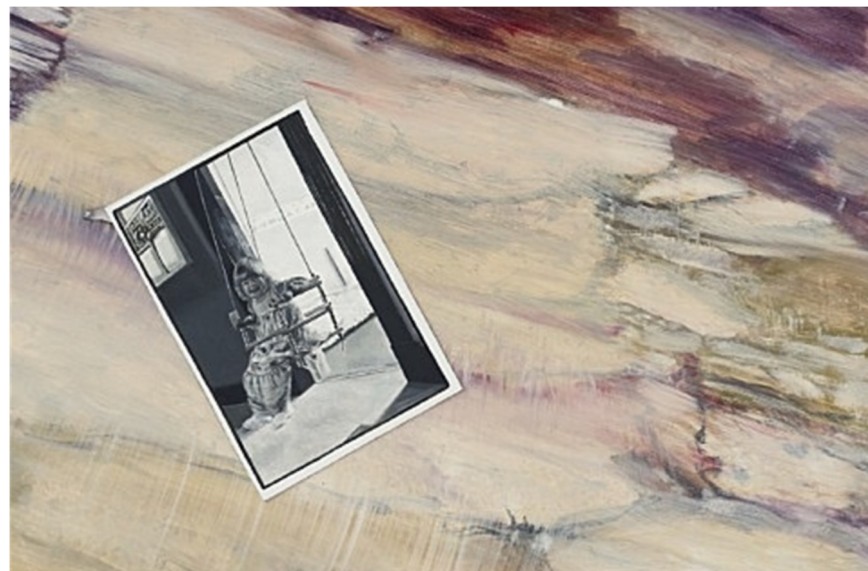

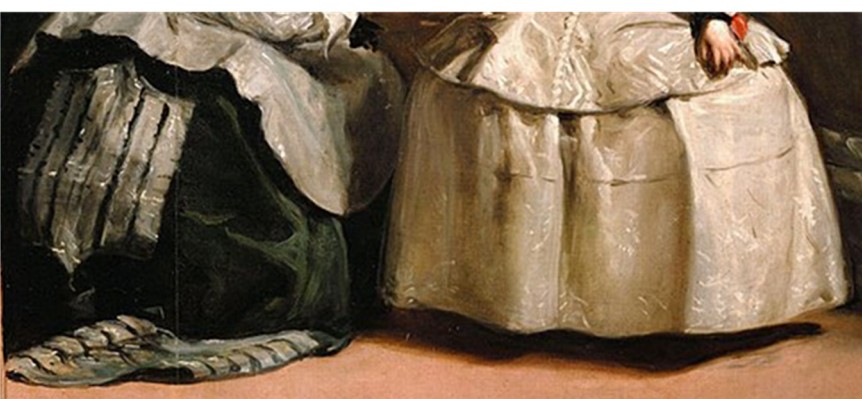

**Figure 4.** Details (**Top**) (McKenzie 2021). (**Bottom**) (Velázquez 1656).

*2.3. The Jar*

The little girl in the centre of the meninas is being handed a little *búcaro* pitcher (Figure 5). A shiny, red vessel, suitably small. It is extravagantly presented on a silver tray, despite having a handle. The ceremony of it all, the protocol, the excess.

What is hardest to understand today, though, is not hierarchy or adulation, but that *búcaros* were *fragrant*. The clay, either because of its own properties, or because of how it was treated, contained aromatic molecules (simultaneously itself, and more than, plural), This particularity added a tiny sensorial thrill to the water they held. Glossy and sweet, tempting stuff.

They went from container to content, as they would be chipped and consumed, slowly, totally (rumour has it, that this form of pica provided the preferred pallor of the time (Grovier 2020)).

> 'Fashionable Spanish ladies were known to eat small fragments of the búcaros to benefit from certain gastronomic qualities. The aroma could be enhanced by storing the absorbent clay vessels in boxes scented with spices and oils.' (Victoria and Albert Museum Editors 2016)

(There is a metaphor, here, somewhere, relevant to autotheoretical practices. About little bodies made of clay, and fashionable paleness, and something to do with Biblical stuff, the Genesis and communion, apples and bread and so on. About being conceived as a consumable, and a fragrant one at that. About the feeling of dry clay between milk teeth, the tiniest crack and its brittle resonance in the skull, the threat of gum blood. About the practice that disappears a whole little jar, morsel by morsel, day by day. Tanto va el cántaro a la fuente, que al final se rompe (so often the pitcher goes to the fountain, that in the end, it breaks). Careful with what you wish for, with whatever pleases you. Not all vessels want to be eaten. Not all origins are muddy.)

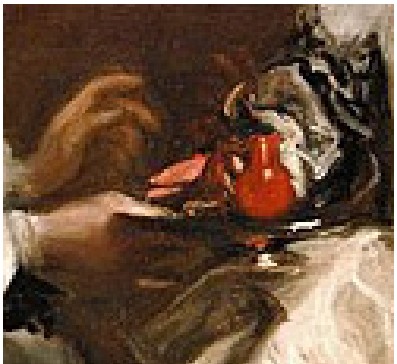

**Figure 5.** Detail, *búcaro* (Velázquez 1656).

### 2.4. The Hole

When I first came across *Diagrammatic Writing*[5], it had the exact shape of a hole I was struggling to feel out, rather than fill up, when it came to discussing autotheory.[6] This hole, namely the way many autotheoretical and autotheoretical-adjacent works seem to play with page layouts as part of their citational performances, felt pressing (as in, it demanded attention, insisted on contact). Like the texts Pollock complains about, these works that grabbed my attention 'carry [their] own *faux* referents: stylish, trendy, clever, avant-garde, projecting in turn a kind of new formalism' (p. 75).

(A potential danger: holes and sore spots tend to invoke fingers or tongues, refusing to be left alone. And the exploitation is not only textual, but personal;[7] and the text stops performing to allow the pain to do so. And textual little tricks may offend, or tire, but public little pathetisms have a way of becoming an honoured guest, less of a taste of fragrant clay, and more of a bite.)
(I am, of course, rather invested in pretentious little tricks.)

Barthes has surely sparked many of these textual and diagrammatic experiments. I am thinking, for instance, of the relationship between *A Lovers' Discourse*, and the margin games in Maggie Nelson's *The Argonauts*. Other theorists, such as Julia Kristeva and her dual columns in 'Stabat Mater', have also played with how to visualise the auto and the theory, the one and the other, the discourse and the meta. I collect contemporary iterations that sprawl into creative hybrids: Nelson, of course, but so many others have played with similar devices, such as Joanna Walsh in *Break.up*, Meena Kandasamy in *Exquisite Cadavers*, or Lisa Robertson in *Anemones-A Simon Weil project*. The example list is, like the entangled

field, constantly expandable and on-going, a *compendium of itself*, being-such *with all of its predicates*, Lovable, as Agamben would say. A selection of whatever pleases.

### 2.5. The Self-Portrait

My introduction to the term quodlibet was via Lucy McKenzie's works, mentioned earlier. McKenzie, who is classically trained in trompe l'oeils, pushes the tradition to the conceptual realm, while preserving all its rigour (and, of course, the terminology). She often plays with simulacras of interior architecture. In her installations you find yourself before commercial murals, all sorts of stone slabs, doors and heating vents that are not really there: surface look-alikes that hold a mix of object look-alikes.

McKenzie's quodlibets, like any trompe l'oeil, play with trickery, with what is there and what is not, what is true, and what isn't, plus the self-awareness these require. But something particularly interesting about her contemporary interpretations is that she often adds a (self)referential, performative tang. There is one in particular, *Quodlibet XXVI* (*self-portrait*), from 2013 (Figure 7, bottom), which appears to be a very simple corkboard, with a cheap pine frame, holding an email from the artist (4 February 2010) printed on an A4, fastened with a single pin. Someone has scratched out the names from the printed email with a pen. It refers to pictures taken of her for an artwork that is now at risk of being used as pornography, and declares that although those are not technically her work, she feels 'entitled to some say' given that it is a picture of her body. '[I]f he uses my image or any reference to me it is against my will. I do not wish to provide his content." (Hendrix 2013, where a picture of the letter is large enough to be read).

This quodlibet is, in a way, about what pleases others, but (rightfully) displeases her. It is a self-portrait, yes, with a myriad of interconnected references, diagramming relations.

(A constant worry my criminal methodology provokes is that of appropriation. In grabbing different materials to build with, where are the limits? Cultural lines? Personal boundaries? I can quote dead authors all I want, but am I somehow mis-using McKenzie's image myself? Am I following academic rules closely enough to justify this use? Will these parenthetical fragments provoke a failure of the whole? Or is my own presence a way of balancing any taking?)

### 2.6. The Example

The self- in the self-portrait, and the auto- in the autotheory are, in a way, forms of the author presenting themselves as a specimen, an example of self: this is what being looks, feels, sounds like (Velázquez standing proudly besides a turned canvas).

(There is a distinct possibility that this is all a reversal into the mirror stage (Figure 6), pointing at a text and saying, oh, look, it's me!)

Agamben makes a special point of discussing the condition of exemplarity, given that the example is the 'one concept that escapes the antinomy of the universal and the particular;' (p. 9) and is thus 'a singular object that presents itself as such, that *shows* its singularity.' (p. 10) There may be trees (since Agamben seems to enjoy discussing trees) that are a group, and there may be individual, particular trees; but an example tree is, counterintuitively, in its own category due to its exemplarity. It is singled out by its position as a demonstrative of something commonly shared among all trees.

Thus, Agamben concludes, 'the proper place of the example is always beside itself, in the empty space in which its undefinable and unforgettable life unfolds. This life is purely linguistic life.' (p. 10) It is easy to imagine the auto- being beside themselves. It echoes the field-changing 2018 paper by Lauren Fournier, 'Sick Women, Sad Girls, and Selfie Theory: Autotheory as Contemporary Feminist Practice', which curates certain trends in autotheoretical self-fashioning. It is also reminiscent, in a different way, of a scene in the

New Testament where the apostle Paul is making a case for himself before the authorities, one of whom, in hearing all he has to say, utters 'Paul, you are beside yourself! Much learning is driving you mad!' (Acts, New King James Version, 26:24)

　　Much learning may, indeed, condemn you to a beside-ness, or to the autotheoretical sickness and sadness. And this linguistic beside-ness, and its corresponding linguistic self, present a unique spatial opportunity to become entangled in a field of diagrammatic writing. The perfect spot for theory and self to meet. For other selves to join in[8].

> (I told you earlier, this text is not about the three little girls. But it may be that it is a playground built for their encounter.)

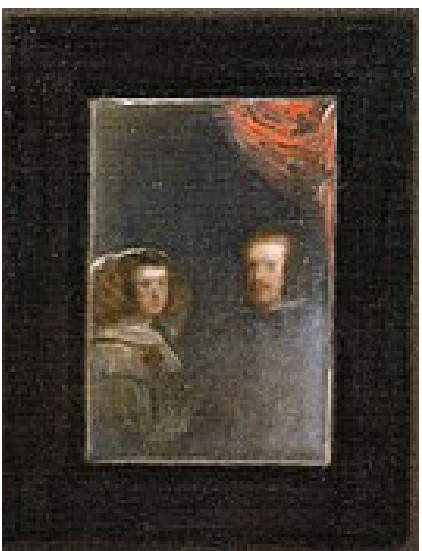

**Figure 6. Detail,** mirror (Velázquez 1656).

### 2.7. The Canvas

　　The Flemish Cornelius Norbertus Gijsbrechts, like Velázquez, ended up being a court painter. In his case, in Copenhagen, in the late 1660s and early 1670s. Gijsbrechts' talents excelled not in portraying people, but objects. He was a trompe l'oeil master, known for his quodlibets, one of which is shown in Figure 4.

　　His oeuvre is obsessively detailed, and his virtuosity is shameless. Every paper creased or folded, every knot and bow tied, the desire to trick the eye into believing them three-dimensional, violent. Some paintings even overcome the limits of a rectangular canvas as he cuts them to the perfect outline of whatever is represented.

　　But one of his most sophisticated works is, unexpectedly, the much (compositionally) simpler *Trompe l'oeil. The Reverse of a Framed Painting*, 1668–1672 (Figure 7, top). It is exactly what it sounds like: a painting that appears to be the back of a framed canvas. The wooden frame has leftover marks of a dark varnish and it surrounds the wooden stretcher of a small canvas. We can see a few millimetres of frayed textile caught between the stretcher and the frame, with nails that barely hold it into place. A red wax seal holds a note that reads '36'.

　　The painter allegedly wanted the piece to be exhibited resting on the floor, frameless—artistic bait (Stoichita 2000, p. 264). Of it, Victor I. Stoichita says in *La invención del cuadro. Arte, artífices y artificios en los orígenes de la pintura europea* (again, the final sentence of a book): 'With this liminar experience, the painting acquires full self-consciousness, of its being, of its nothing.' (p. 266, own translation from the Spanish translation).

(If you look at the facial expression of the girl wearing only knickers, that may be what you see, the acquisition of full self-consciousness, of being, of her nothing.

If you are hearing Morris' box go off, you are not the only one.)

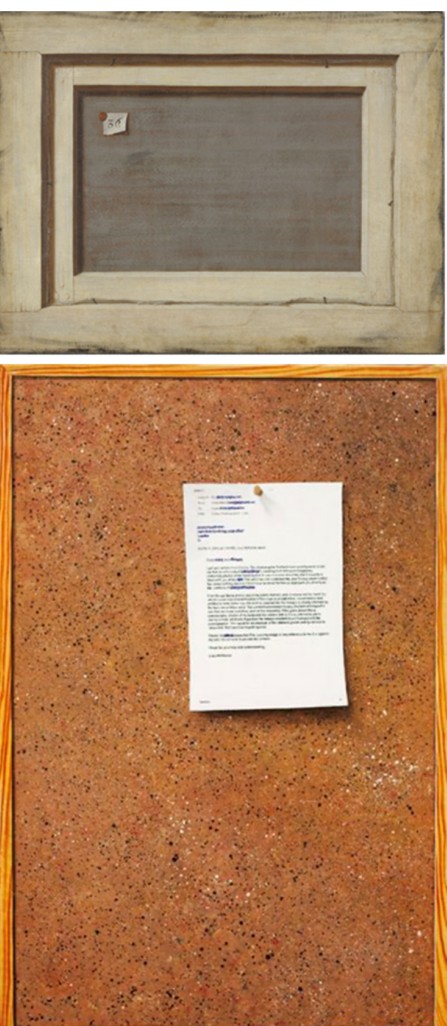

**Figure 7.** (**Top**) *Trompe l'oeil. The Reverse of a Framed Painting* (Gijsbrechts 1670). (**Bottom**) *Quodlibet XXVI (self-portrait)* (McKenzie 2013).

### 2.8. The Easel

A more recent quodlibet by McKenzie, *Quodlibet LXXI*, was part of a series where large trompe l'oeils were presented vertically, as if on an easel, supported by two stilts that bolted to both floor and ceiling (Figure 8, left). Each of these larger, vertical pieces had a counterpart, displayed almost perpendicularly, like an angled desk, right below them. The larger, vertical paintings were replicas of actual, three-dimensional artworks the artist grew up with. And each attached 'table', a quodlibet, featuring depictions of family pictures where the former appeared:

> Her parents, Ray and Lorraine, decorated their family home with art works they bought or were gifted by students of the Glasgow School of Art in the 1970s and '80s. Three of these, a trio of low-relief, abstract wall sculptures, are recreated for *No Motive* as trompe l'oeil, to scale oil paintings. ... These three abstracts are shown freestanding above 'quodlibets', tabletop trompe l'oeil compositions

depicting family snapshots of the corresponding artworks in domestic situ, both haphazard (strung with Christmas cards) and more staged (baby photo-sessions). The quodlibets are intended to function as oblique, mute information boards to the works they accompany. (Galerie Buchholz 2021)

*Quodlibet LXXI*, in particular, was displayed beneath *Beige textiles (attributed to Joseph Linley)*. *Beige textiles* looks, as the name indicates, rather plain, not unlike *The back of a canvas*. There is (the illusion of) a wooden grid, on which scraps of light colored textiles are either stretched to meet the wood, or balled up and stuffed within it.

*Quodlibet LXXI*, on the other hand, is made to look like a multicoloured slab of stone (soapy green and deep purple). On it, instead of a collection of items, a single picture (like in the self-portrait, where there was a single sheet of paper): a little girl (which I guess is the artist herself), wearing a voluminous dress, sitting in a swing chair that seems to be hung from the threshold of a door, flying in the light beam, looking brightly towards the camera as the original artwork hangs to her right.

The concept of 'pleasing', in this installation, may be close to Agamben's, a grab towards the subject's 'own taking-place—toward the Idea.' A subject that is exemplary. She is beside herself, and beside others artists' work as they become hers in these performative paintings.

> (Imagine this parenthetical as a slanted desk in multicoloured faux stone.)

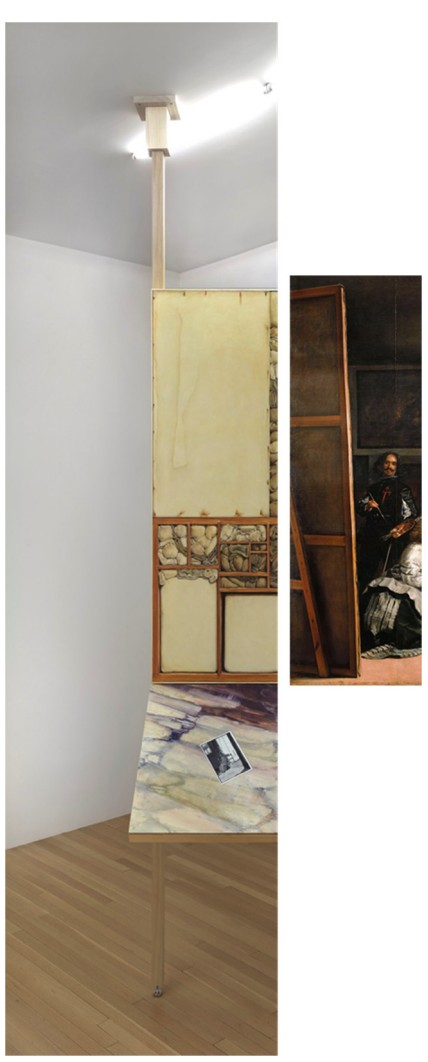

**Figure 8.** Details, canvases (**Left**) (McKenzie 2021). (**Right**) (Velázquez 1656).

## 3. The Linguist

### 3.1. The Textuality

Estrella de Diego writes 'Representing Representation. Reading *Las Meninas*, Again' on the back of Foucault's *The Order of Things: An Archeology of Human Sciences*. For de Diego, who is Spanish herself,[9] the painting is 'a means to materialise our cultural "selves"' (p. 151) to the point where 'it represents "us,"' (p. 151) and we 'belong to the surface of the painting.' (p. 151) This representation is less illustrative (since we are not literally there) than diagrammatic, a dynamic relation.

Earlier, I suggested that *Las Meninas* is Velázquez's self-portrait, but also a group portrait. The two converge: it can be read as a self-portrait as a multitude. In the painting, Velázquez is not only a figure; but also the court where he belongs. The crowd in the image justify him, *make* him: without a court to paint, there would be no court painter. Relations and belonging are key. Adding de Diego's thought to this logical sequence, the plural self-portrait breaks open: it represents 'us' because we belong to 'it', and vice versa Because we have cultural and personal ties with this piece, we can claim it as our own, and as ourselves. Which suits the whole perception 'game' between the beholder and the painted mirror, and the king and queen reflected in it (are they the models, standing where the beholder stands ('I am the king; I am the queen; I am the king and the queen; I am me again' (Snyder 1985, p. 543)); or are they the painted reflection of the work in progress on a turned canvas we cannot see (Snyder 1985, p. 547)?).

These issues do not bother de Diego. Her text builds up the, again, Foucauldian argument by which the palace's room becomes text-like: 'a space in which the Same and the Other can comfortably meet . . . That space is, no doubt, a linguistic space.' (p. 155) She doubles down, never forgetting she is talking about a painting, and thus integrating the visual aspect into the case:

> The visual space becomes a kind of textual, linguistic space, the place where the conflict manifests itself. The conflict is not painterly at all, not even narrative, but linguistic, as it implies subjectivity, the very formation of the Self and the Other; to antithetical, paradoxical, condemned-to-separation concepts that bump into one another in Velázquez'a painting. (p. 157)

> (Drucker's gadflies and diagrammatic entanglements bumping within the page. If that produces any sound, we cannot hear it.)

### 3.2. The Recap

> (Life, according to Agamben, is linguistic.
> The palace, according to Foucault and de Diego, is language, a linguistic place.
> Velázquez, in the painting, can only be language, too - like you, like me, linguistic being/s.
> I am not the king nor the queen, and most definitely not Margarita.
> I am the way hands flutter in the air as the characters speak, 'suspended' (de Diego 2003, p. 154), quiet.
> I am a borrowing that turns out to be more of a breaking.
> A stubborn window without blinds.
> A criminal flicker of pleasure.
> This text is a board, a table—an interface.
> A ground for a little girl I barely remember being, but who spent enough time staring at the art encyclopedia to make painted people, long time dead, her playmates.)

*3.3. The Citational*

In this playground, citations have performative capabilities. Capabilities such as 'us[ing] language like paint' (Pollock 1998, p. 80), in the sense of invoking images, rendering the absent, present. For instance, I can tell you (as in, I can invoke in your mind) that the two paintings in the background of the palatial room are reproductions of famous pieces that Velázquez found relevant enough to include in the *Meninas*.

The effort of painting these other paintings within the *Meninas* is calculated; replicating replicas, a form of visual quotation it itself, really. The gesture of citing, which is a slightly different thing from the citation itself, is performative, too. In writing, it alters the genre of the text (which can displace it to an in-between, a hybrid, like autotheory: 'the threat of breaching the boundary is visibly present.' (Drucker 2013, p. 20)). It expands its universe by becoming familiar, meaning it traces a genealogy with those texts that employ certain textual devices as a self-aware quirk (which, in turn, could be considered the reverse from the example above, where paintings become citations, here, instead, a metamorphosis from text to *diagram*, a relation).

And yet, what I want to play with is the idea—speculative, to be sure -that it is precisely this citational gesture that can turn something into theory. If I type the content of a painted letter to make a point about self-referential material in quodlibets, it is digested into the discourse, it *becomes* the discourse. Same with little hands clipped from a larger painting, making a wordless point (but not language-less, as we have just seen). The moon-landing of materials is as revelatory of the self as the autobiographical bits, if not more. It is, unlike a classical academic essay, an unveiling of whatever pleases someone, and that turns out be a pretty intimate portrait.

Auto-theory is not only the combination of the self ('the body, and other so-called personal and explicitly subjective models' remembering Fournier's definition (p. 7)), and academic discourse; but a way to legitimise this subjective model, this self, *as* theory— something 'to consider, speculate, look at' (Harper).

And thus I speculate that *my* gestures, which are ultimately an expression of embodied experience in this becoming of theory, can be theory, too.

> (I am straying towards boxes that make sounds again,
> nervously pinching myself.)

The citational gesture is a performative act that effectively alters the quality of the material being cited, at least in the context of the work into which it is being introduced, like a *búcaro*, augmented clay, content and container, surface and depth. The citational gesture makes any material something 'to consider,' and thus, theory. It is also able to charge the relational space between text fragments, spreading its power on to the page and its space. This evidences what Drucker already knew, that 'the space between is not neutral' (p. 12).

> (That space, too, is theory, here.)

In the space of the palace, everyone has become a citation, a sampled subjectivity ('Velázquez presents himself in this work as contemplating the subjects of his painting, reflecting on something more fundamental than the way they look: he is thinking, of course, about the way they should be represented.' (Snyder 1985, p. 563)). They are linguistic creatures whose allure is still strong today, demanding contemplation. You cannot help but to obey them; and in doing so, feel like you see yourself, because of the mirror, because of the lack of fourth wall, because this representation escapes the illustrative into the relational because, as de Diego says, we belong on their surface.

> (Which is to say their skin can be your skin.)

Which might make them, and us (everyone!) theory, too. Autotheory ('The world is perceived by the way it's named' (de Diego 2003, p. 150)). Self-performing tricks that reveal an entangled subjectivity through time in a small, linguistic space.

### 3.4. The Mirror

I have been sneaking quotes here and there by Joel Snyder. They come from a paper called '"Las Meninas" and the Mirror of the Prince.' If you crave schematic illustrations of the palace room, the article is going to delight you. They abound: with the canvas, without it, from where the margin of the painting ends, from further behind that non-existing fourth wall. The big revelation, though, is not to be found in those sketches, but in the way Snyder proposes that the mirror is a pun (p. 559). And a linguistic one at that, too.

For Snyder, it refers to a type of literature popular at the time, *Speculum principis* or 'the mirror of the prince.' (p. 558) These types of books were 'an essential part of the process of self-fabrication . . . manuals of self-fashioning that provide maxims for the construction of the self according to exemplary standards. These books themselves are works of art that provide guidance for the production of a work of art—a work of art that is the self itself.' (p. 558)

So, we have an image that is a reference to a type of text. And this type of text is meant to educate on the ideal self, one's own self-fashioning, to the point where it may become a form of art (and the pun gets out of hand, from having shifted so often from text to image, and so forth).

(The overworked pitcher hitting the base of the fountain.)

This notion of the exemplary, here, refers to 'a reflection that can be reached only through art and can be seen only by the inner eye.' (p. 558) Which seems something performative writing could do, language like paint, and all that. This exemplarity is also beside itself, and the empty space it occupies is the metaphorical/painterly mirror. What we see in the painting is the blurry ideal of its intended audience, the monarchs. And in posterity, an appeal to a heightened self of whoever is watching (you, me, us, whoever belongs to its surface: 'This mirror can reflect only images existing in and through art.' (p. 559))

'It is an image of character, disposition, thought', writes Snyder (1985, p. 559). 'An image whose source is the imagination and whose cause is in art.' (p. 559) I interpret this as a form of performative quotation of the inner self, rendering the invisible, visible, augmented, bettered, 'fashioned by art, in accord with divine doctrine and the wisdom of men of arts, letters and practical affairs,' (p. 558) not unlike the well-read, emotionally available subjects of autotheory.

### 3.5. The Body

The turned painting haunts my intentions. It is not that I need to know once and for all whether the reflection in the mirror is of its surface. It's that I wanted it to be a door. Not a literal one, there is already one of those, in this image (Figure 9). I wanted it to be something like a portal towards the possibilities of indirect biographical narration in autotheory (Stoichita writes that 'The reproduction in painting of real openings can be interpreted as an autobiographical confession of an image' (p. 59), which is fantastically obscure and evocative. I imagine he means that they are an opportunity to reveal something, a breach). What I had in mind as an example is a daring and compelling book, Jenny Boully's *The Body*. A text literally beside itself, since it is composed of pages that are blank (no body of text), except for footnotes that force the reader to grasp for meaning, and imagine what they refer to.

(Alas, I am more mismanaged window than turned canvas. No mystery, just compulsion.)

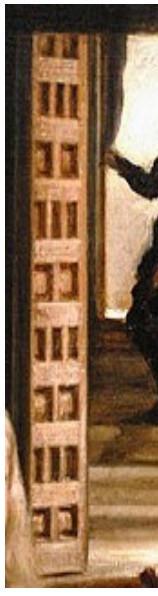

**Figure 9.** *Las Meninas*, door detail (See Figure 1).

How could a first-person voice imitate the canvas' position and turn its back to the reader, but remain in the scene? There is, of course, plenty of theory without auto-. What I wanted to look for something that crossed the threshold,

(a swing hung from a doorframe)

an auto- that was palpable but not a core. A pulse, almost intrusive,

(the gadfly/pebble)

a tease of personhood that does not fully cooperate with the self-fashioning and giving away of the self, does not unveil a wound to tempt a finger,

(or a whole fist)

stubbornly present, but not quite productive.

And why? Why go to such lengths for a voice that refuses to contribute? How does one justify such presence, when the key aspect of the autotheoretical self is its usefulness in nuancing and embodying otherwise much more abstract discourse? Maybe, like the small boys on the right in both the *Meninas* and my childhood picture, this voice is simply an expression of being: being-in-motion[10], being-as-becoming, being-as-language:

(language in motion as being)

alien to productivist pressure, unable or unwilling to maximize the impact of its presence, opting instead to appear as a gestural affectation, a hint of a body others can recognize and hold on to.

(and maybe even side with, hands running a parallel discourse to words)

*3.6. The Figure*

(The other way to see these is what Barthes calls the Moussu trope[11] a form of 'truth' disguised as a subordinate clause. Not any parenthetical qualifies as such, of course. Barthes calls these 'a full-fledged rhetorical figure':

'I often have to put things I consider very important in subordinate clauses or in parentheses, I consider it to be a full-fledged rhetorical figure to which I've given a proper name . . . There was an extremely respectable woman, gentle and

a bit intrusive, ... her name was Madame Moussu, ... At one moment, when Madame Moussu, whom I didn't know, saw me light a cigarette, she said to me: "oh, my son always says: Since I began at the Polytechnic, I stopped smoking." There's a rhetorical figure in which the principal and only information, namely that her son was a polytechnician, was given through a subordinate clause. If you notice present-day language, we all do that. It's thus a true rhetorical figure.'

(Barthes 2005, p. 223)

In comparison to this Barthesian figure, these handsy little textual incursions, despite also being 'gentle and a bit intrusive,' are less of a humble brag (the contemporary sibling of the Moussu troupe). That said, they are still a confession that the subordinate auto- is flawed, mostly uninteresting, with baggage and biases, unable to steal lip glosses and similarly weary about appropriation. This unproductive voice is (on a good day, when things work according to plan) a sub-clause of the collective discourse.

Any Moussu-ness to be found between these parenthesis would ideally be bothersome enough to become a Druckenian gadfly (Moussu to *mosca*), capable of 'goading' critical thought. But, instead, I'll take the provocation of a haptic reaction as a victory. What I mean by haptic is what Laura Marks[12] describes: a form of 'touching, not mastering' (Marks 2002, p. xii) that comes from an immediate engagement 'with objects and ideas and teasing out the connections immanent to them' (xiii). A haptic critic sticks to surfaces, plays with mimesis, 'getting close enough to the other thing to become it' (p. xiii). Trompe l'oeil, representation, self-awareness, canvas, and ultimately, skin.

Sigh.

All these notes always lead to skin. Like when you sit down in a buttoned shirt, and there is the tiniest belly flash. A tension (attention!). A bit embarrassing, a bit normal, a bit navel-y. The stubbornness of skin to be present as both surface and method (crash!), be it via thigh pinch, a gadfly, or other devices.)

### 3.7. The Intimacy

Diagrammatic writing, we established, reveals relations internal to the text, and possibly external ones, too[13]. And performative writing, as a 'material practice' (Pollock 1998, p. 86) can embody a subjectivity understood 'as the performed relation between or among subjects.' (p. 86) This notion of subjectivity as a performed relation circles right back to the diagrammatic ability, but also to the palace where Velázquez stands, plural, like the beholder ('I am the king, the queen'). So the subjectivity can also be diagrammatic, a plottable study of relations, and fancies, as it 'tends to *subject* the reader to the writer's reflexivity, drawing their respective subject-selves reciprocally and simultaneously into critical "intimacy."' (Pollock, p. 86)

In the intimacy of this text, for instance, there has been less of a conceptual red thread, and more of a lattice made with old-fashioned red ribbon. This text has been a background as well as a pin-board stuffed with letters and books and quills. A surface where the holding line is fanciful, and creates crossings like bows or rosettes. These guide the eye, fabric details in a dark Spanish room, fastening and decorating. Practical in the providing of some sort of structure, but also, why pretend otherwise, a frivolous whimsy.

Red clay that may contain water, at the same time as flavour it, and in doing so, volunteer itself for a bite.

(All hands mid-air, waiting for the swing to fly back, waiting for a moment when the light moves, when they can push, finally touch flesh, their own, always, since when they reach out and grab the world past the surface, pull the red thread (Figures 10 and 11), they unveil the many behind the curtain, all there, 'suspended', quoted and reflected, being-such, Lovables, loved, beside themselves, being me, being you, being all of us, in a dark room that holds the sound of its own making, gesticulating, staring at an audience that is not there.)

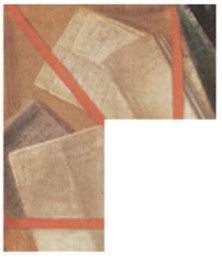

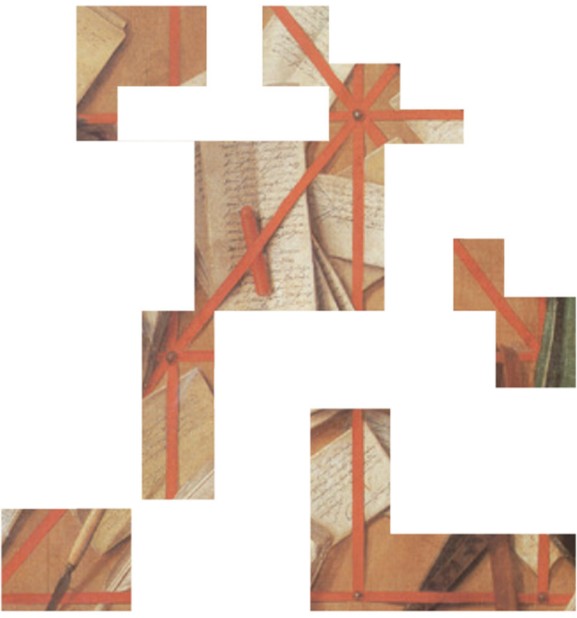

**Figure 10.** Red ribbon, detail (Gijsbrechts 1664, see Figure 5).

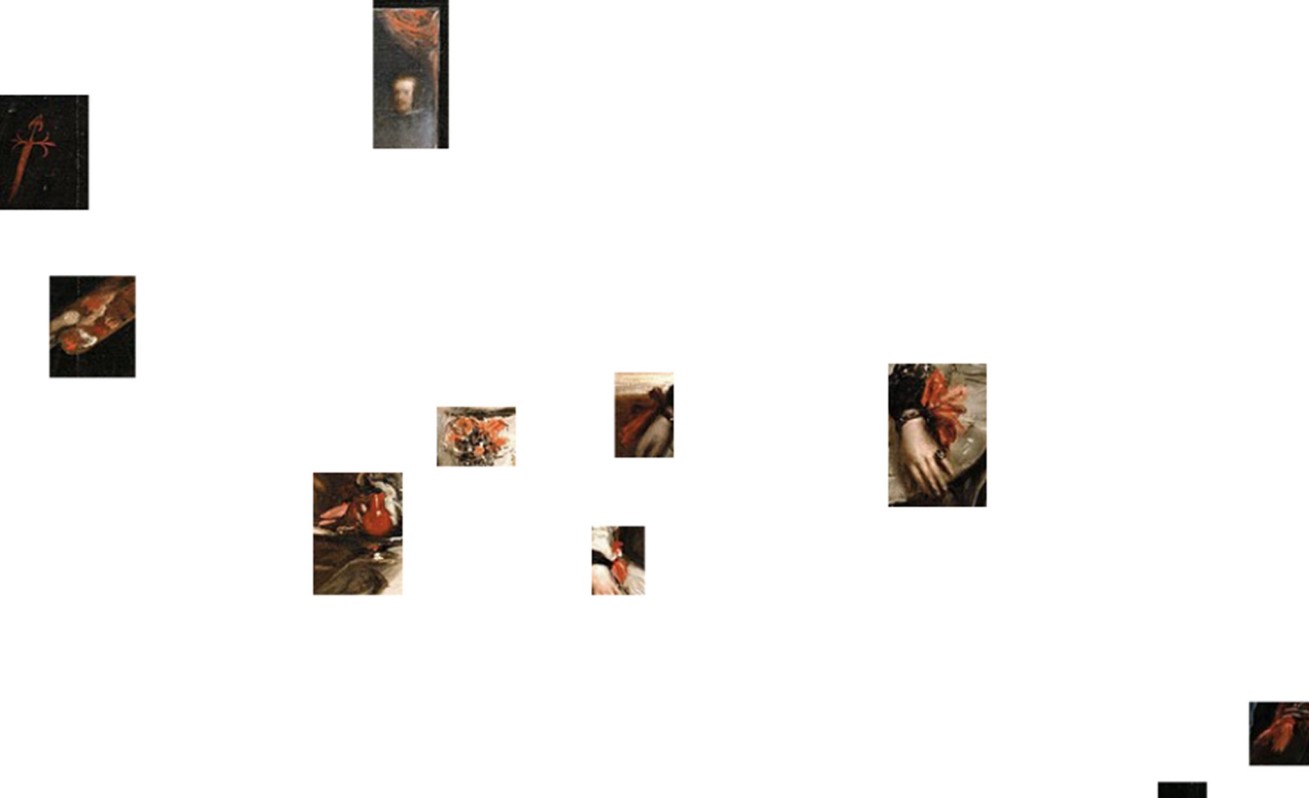

**Figure 11.** Red, details (Velázquez 1656).

**Funding:** This research received no external funding.

**Institutional Review Board Statement:** Not Applicable.

**Informed Consent Statement:** Not Applicable.

**Data Availability Statement:** Not Applicable.

**Acknowledgments:** Heartfelt thanks to the academic editors of this issue, Katherine Baxter and Cat Auburn for their support.

**Conflicts of Interest:** The author declares no conflict of interest.

## Notes

[1]     I would like to thank Sarah Jackson, Delphine Grass, Helena Hunter, and Hannah van Hove for letting me think out loud with them through this and other tangles in this text; and for making and holding a shared space for writing.

[2]     Augmentation is a roughly pillaged term from Lisa Robertson's *The Baudelaire Fractal*, 2020: 'To augment would be my work—to add the life of a girl without substracting anything else from the composition, and then to watch the centre dissolve. It is exactly this sense of augmentation, which is to say, not necessarily an expansion or enlargement, but a timely complexification, sometimes an argumentation, at others a dissolution or the invention of a new form of refusal, that makes the poem a possible space.' (p. 141)

[3]     The idea of alunizaje as an artistic research methodology came up early in my doctoral research, in conversations with my supervisor Kris Pint, and it has proved fantastically generative since then.

[4]     I was reminded of Morris' box when I read Amy McCauley's *Propositions*, 2020: 'Forms of art propose an ethics of being. They achieve this by containing the sound of their making' (p. 13).

[5]     in Anne M. Royston. 2019. *Material Noise: Reading Theory as Artist's Book*. Cambridge, MA: MIT Press.

[6]     It is worth mentioning Robyn Wiegman's 'In the Margin with the Argonauts'. 2018. *Angelaki*, vol. 23, n1, Feb 2018, an article that *does* tackle some of this issues, as the title indicates, in Maggie Nelson's *The Argonauts*. Nonetheless, the emphasis of the texts falls more on authenticity and media. I owe this reference to Cat Auburn. Additionally, I want to note two other papers that I only came across much too late to include in this text, and do touch on formal experimentation in autotheory and point in similar directions to what I am attempting here. One is Sarah Minor's 'Looking While Reading I, II, III', in the *Journal of Creative*

*Writing Studies*, 2021 ('Auto-theory . . . is theory, embodied; an engagement with literary discourse that is consistently interrupted by subjectivity, by reminders of the physical body writing from a specific context. By rejecting both traditional boundaries and linear progression, what is formal in auto-theory communicates conceptual and political aims.' (p. 11). The other is Carolyn Laubender's 'Speak for Your Self: Psychoanalysis, Autotheory, and The Plural Self', in *Arizona Quarterly*, 2020 (' . . . Nelson constructs 'a plural self' . . . as a literary avatar designed to pay homage to the inter-relational contours of the self. Nelson's much-noted formalist innovations, including textual italicizations and marginal citations, strategically craft a narrative self that is not so much undone as it is remade, pluralized through its multiple textual incorporations of the other.' (p. 41)).

7    I kept Hannah Gadsby's *Nanette*, 2018 (the infamous stand up show where she paradoxically announces she needs to quit comedy, since she relies on making herself the butt of the jokes, which is hurtful to herself and those who identify with her) in the back of my mind as I wrote this piece, but did not mention it at first. While waiting for the reviews, I found myself outside a London pub, after a conference, enthusiastically squealing at Audrick D'Mello, as we both agreed that Gadsby's show was autotheoretical. I owe him the realization that this reference may be useful to others.

8    Adriana Cavarero's *Inclinations: A Critique of Rectitude*, 2016, is a far away reference to this thought, too.

9    Estrella de Diego was my (neeedless to say, absolutely fabulous) MA thesis supervisor, but that was many years ago, and she had no involvement in this text.

10    This is a nod to Aby Warburg's *bewegtes Leben*, which translates as life in motion, which is how he described the movement depicted in the still images he researched.

11    Another note of debt for Kris Pint, who first introduced me to this Barthesian scene, and then, when I needed it, reminded me of where to find it.

12    I would not have come across Mark's work if it was not for Arne de Winde.

13    Some of what I have attempted to do in these footnotes is to plot the relationality of thinking, reading, and writing. I am trying to move from the solitary auto- to a group portrait, even if they remain para-textual attachments. This may, in turn, borrowing from Drucker's lecture, affect the performative materiality of the footnote, and its semantic value.

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
