# Peer review of "Quodlibet with Meninas"

_arts, 2022_

Round 1

Reviewer 1 Report

It presents an original theme, based on diagrammatic writing/imagery in analogy to the painting by Velasquez Las Meninas.

However, it lacks: a history of the diagrammatic in art and an  introduction to the methodology used in the text. Without these elements the text becomes unclear.

Author Response

Dear Reviewer,

thanks so much for the time spent with my piece.

I want to start by explaining that I had not realised the upload system would alter the formatting of my piece. The original layout played out part of the performativity of the piece and was part of the discourse. Without it, it was definitely confusing.

Thanks for still engaging with the work in a version that, I am afraid, made significantly less sense.

I hope part of the issues you rightly pointed out were addressed by restoring these features. That said, I have also added a new section, 'The codex', that delves deeper into Drucker's work on diagrammatic and their performative implications.

I have also tried addressing methodological issues with tiny incursions that point out how the text is written and how the layout plays a role. 

I appreciate your helpful suggestions so very much. Thank you.

Reviewer 2 Report

Very interesting and also personal, almost intimate paper which also has very stimulating visual qualities.

On the following lines I had the impression that either words and spaces or character size should be checked/corrected: 185 (correct 'compedium'), 194 (involuntary text insert?), 278 and passim (correct Mckenzie -> McKenzie), 294 (spaces?), 539-543 (lines, character size, inserts?).

Author Response

Dear Reviewer,

thanks so much for the time spent with my piece, and for such a kind, careful reception.

Your observations on a possible copy/paste snafu made me realise that the upload system altered the formatting of the original piece. The layout played out part of the performativity of the piece and was part of the discourse. Without it, it was much more confusing.

Thanks for still engaging with the work in a version that, I am afraid, made significantly less sense.

I have therefore restored the design, and addressed all the typos you identified. 

I appreciate your helpful suggestions and gentle corrections so very much. Thank you.